# Stochastic Mirror Descent in Variationally Coherent Optimization Problems

**Zhengyuan Zhou**
Stanford University
zyzhou@stanford.edu

**Panayotis Mertikopoulos**
Univ. Grenoble Alpes, CNRS, Inria, LIG
panayotis.mertikopoulos@imag.fr

**Nicholas Bambos**
Stanford University
bambos@stanford.edu

**Stephen Boyd**
Stanford University
boyd@stanford.edu

**Peter Glynn**
Stanford University
glynn@stanford.edu

## Abstract

In this paper, we examine a class of non-convex stochastic optimization problems which we call *variationally coherent*, and which properly includes pseudo-/quasi-convex and star-convex optimization problems. To solve such problems, we focus on the widely used stochastic mirror descent (SMD) family of algorithms (which contains stochastic gradient descent as a special case), and we show that the last iterate of SMD converges to the problem's solution set with probability 1. This result contributes to the landscape of non-convex stochastic optimization by clarifying that neither pseudo-/quasi-convexity nor star-convexity is essential for (almost sure) global convergence; rather, variational coherence, a much weaker requirement, suffices. Characterization of convergence rates for the subclass of strongly variationally coherent optimization problems as well as simulation results are also presented.

## 1   Introduction

The stochastic mirror descent (SMD) method and its variants[1, 7, 8] is arguably one of the most widely used family of algorithms in stochastic optimization – convex and non-convex alike. Starting with the orginal work of [16], the convergence of SMD has been studied extensively in the context of convex programming (both stochastic and deterministic), saddle-point problems, and monotone variational inequalities. Some of the most important contributions in this domain are due to Nemirovski et al. [15], Nesterov [18] and Xiao [23], who provided tight convergence bounds for the ergodic average of SMD in stochastic/online convex programs, variational inequalities, and saddle-point problems. These results were further boosted by recent work on extra-gradient variants of the algorithm [11, 17], and the ergodic relaxation of [8] where the independence assumption on the gradient samples is relaxed and is replaced by a mixing distribution that converges in probability to a well-defined limit.

However, all these works focus exclusively on the algorithm's ergodic average (also known as time-average), a mode of convergence which is strictly weaker than the convergence of the algorithm's last iterate. In addition, most of the analysis focuses on establishing convergence "in expectation" and then leveraging sophisticated martingale concentration inequalities to derive "large deviations" results that hold true with high probability. Last (but certainly not least), the convexity of the objective plays a crucial role: thanks to the monotonicity of the gradient, it is possible to exploit regret-like bounds and transform them to explicit convergence rates.[1]

By contrast, the gradient operator of the non-convex programs studied in this paper does *not* satisfy any reasonable monotonicity property (such as quasi-/pseudo-monotonicity, monotonicity-plus, or any of the standard variants encountered in the theory of variational inequalities [9]. Furthermore, given that there is no inherent averaging in the algorithm's last iterate, it is not possible to employ a regret-based analysis such as the one yielding convergence in convex programs. Instead, to establish convergence, we use the stochastic approximation method of Benaïm and Hirsch [2, 3] to compare the evolution of the SMD iterates to the flow of a mean, underlying dynamical system.[2] By a judicious application of martingale limit theory, we then exploit variational coherence to show that the last iterate of SMD converges with probability 1, recovering in the process a large part of the convergence analysis of the works mentioned above.

**Our Contributions.** We consider a class of non-convex optimization problems, which we call *variationally coherent* and which strictly includes convex, pseudo/quasi-convex and star-convex optimization problems. For this class of optimization problems, we show that the last iterate of SMD with probability 1 to a global minimum under i.i.d. gradient samples. To the best of our knowledge, this strong convergence guarantee (almost sure of the last iterate of SMD) is not known even for stochastic convex problems. As such, this results contributes to the landscape of non-convex stochastic optimization by making clear that neither pseudo-/quasi-convexity nor star-convexity is essential for global convergence; rather, variational coherence, a much weaker requirement, suffices.

Our analysis leverages the Lyapunov properties of the Fenchel coupling [14], a primal-dual divergence measure that quantifies the distance between primal (decision) variables and dual (gradient) variables, and which serves as an energy function to establish recurrence of SMD (Theorem 3.4). Building on this recurrence, we consider an ordinary differential equation (ODE) approximation of the SMD scheme and, drawing on various results from the theory of stochastic approximation and variational analysis, we connect the solution of this ODE to the last iterate of SMD. In so doing, we establish the algorithm's convergence with probability 1 from any initial condition (Thereom 4.4) and, to complete the circle, we also provide a convergence rate estimate for the subclass of strongly variationally coherent optimization problems.

Importantly, although the ODE approximation of discrete-time Robbins–Monro algorithms has been widely studied in control and stochastic optimization [10, 13], converting the convergence guarantees of the ODE solution back to the discrete-time process is a fairly subtle affair that must be done on an case-by-case basis. Further, even if such conversion goes through, the results typically have the nature of convergence-in-distribution: almost sure convergence is much harder to obtain [5].

## 2 Setup and Preliminaries

Let $\mathcal{X}$ be a convex compact subset of a $d$-dimensional real space $\mathcal{V}$ with norm $\|\cdot\|$. Throughout this paper, we focus on the stochastic optimization problem

$$\begin{aligned} \text{minimize} \quad & g(x), \\ \text{subject to} \quad & x \in \mathcal{X}, \end{aligned} \tag{Opt}$$

where the objective function $g \colon \mathcal{X} \to \mathbb{R}$ is of the form

$$g(x) = \mathbb{E}[G(x;\xi)] \tag{2.1}$$

for some random function $G \colon \mathcal{X} \times \Xi \to \mathbb{R}$ defined on an underlying (complete) probability space $(\Xi, \mathcal{F}, \mathbb{P})$. We make the following assumptions regarding (Opt):

**Assumption 1.** $G(x, \xi)$ is continuously differentiable in $x$ for almost all $\xi \in \Xi$.

**Assumption 2.** $\nabla G(x;\xi)$ has bounded second moments and is Lipschitz continuous in the mean: $\mathbb{E}[\|\nabla G(x;\xi)\|_*^2] < \infty$ for all $x \in \mathcal{X}$ and $\mathbb{E}[\nabla G(x;\xi)]$ is Lipschitz on $\mathcal{X}$.[3]

Assumption 1 is a token regularity assumption which can be relaxed to account for nonsmooth objectives by using subgradient devices (as opposed to gradients). However, this would make

the presentation significantly more cumbersome, so we stick with smooth objectives throughout. Assumption 2 is also standard in the stochastic optimization literature: it holds trivially if $\nabla G$ is uniformly Lipschitz (another commonly used condition) and, by the dominated convergence theorem, it further implies that $g$ is smooth and $\nabla g(x) = \nabla \mathbb{E}[G(x;\xi)] = \mathbb{E}[\nabla G(x;\xi)]$ is Lipschitz continuous. As a result, the solution set

$$\mathcal{X}^* = \arg\min g \tag{2.2}$$

of (Opt) is closed and nonempty (by the compactness of $\mathcal{X}$ and the continuity of $g$).

*Remark* 2.1. An important special case of (Opt) is when $G(x;\xi) = g(x) + \langle \zeta, x \rangle$ for some $\mathcal{V}^*$-valued random vector $\zeta$ such that $\mathbb{E}[\zeta] = 0$ and $\mathbb{E}[\|\zeta\|_*^2] < \infty$. This gives $\nabla G(x;\xi) = \nabla g(x) + \zeta$, so (Opt) can also be seen as a model for deterministic optimization problems with noisy gradient observations.

## 2.1 Variational Coherence

With all this at hand, we now define the class of *variationally coherent optimization problems:*

**Definition 2.1.** We say that (Opt) is *variationally coherent* if

$$\langle \nabla g(x), x - x^* \rangle \geq 0 \quad \text{for all } x \in \mathcal{X}, x^* \in \mathcal{X}^*, \tag{VC}$$

with equality if and only if $x \in \mathcal{X}^*$.

*Remark* 2.2. (VC) can be interpreted in two ways. First, as stated, it is a non-random condition for $g$, so it applies equally well to *deterministic* optimization problems (with or without noisy gradient observations). Alternatively, by the dominated convergence theorem, (VC) can be written as:

$$\mathbb{E}[\langle \nabla G(x;\xi), x - x^* \rangle] \geq 0. \tag{2.3}$$

In this form, it can be interpreted as saying that $G$ is variationally coherent "on average", without any individual realization thereof satisfying (VC).

*Remark* 2.3. Importantly, (VC) does not have to be stated in terms of the solution set of (Opt). Indeed, assume that $\mathcal{C}$ is a nonempty subset of $\mathcal{X}$ such that

$$\langle \nabla g(x), x - p \rangle \geq 0 \quad \text{for all } x \in \mathcal{X}, p \in \mathcal{C}, \tag{2.4}$$

with equality if and only if $x \in \mathcal{C}$. Then, as the next lemma (see appendix) indicates, $\mathcal{C} = \arg\min g$:

**Lemma 2.2.** *Suppose that* (2.4) *holds for some nonempty subset $\mathcal{C}$ of $\mathcal{X}$. Then $\mathcal{C}$ is closed, convex, and it consists precisely of the global minimizers of $g$.*

**Corollary 2.3.** *If* (Opt) *is variationally coherent,* $\arg\min g$ *is convex and compact.*

*Remark* 2.4. All the results given in this paper also carry through for $\lambda$-variationally coherent optimization problems, a further generalization of variational coherence. More precisely, we say that (Opt) is $\lambda$-*variationally coherent* if there exists a (component-wise) positive vector $\lambda \in \mathbb{R}^d$ such that

$$\sum_{i=1}^{d} \lambda_i \frac{\partial g}{\partial x_i}(x_i - x_i^*) \geq 0 \quad \text{for all } x \in \mathcal{X}, x^* \in \mathcal{X}^*, \tag{2.5}$$

with equality if and only if $x \in \mathcal{X}^*$. For simplicity, our analysis will be carried out in the "vanilla" variational coherence framework, but one should keep in mind that the results to following also hold for $\lambda$-coherent problems.

## 2.2 Examples of Variational Coherence

*Example* 2.1 (Convex programs). If $g$ is convex, $\nabla g$ is a monotone operator [19], i.e.

$$\langle \nabla g(x) - \nabla g(x'), x - x' \rangle \geq 0 \quad \text{for all } x, x' \in \mathcal{X}. \tag{2.6}$$

By the first-order optimality conditions for $g$, we have $\langle g(x^*), x - x^* \rangle \geq 0$ for all $x \in \mathcal{X}$. Hence, by monotonicity, we get

$$\langle \nabla g(x), x - x^* \rangle \geq \langle \nabla g(x^*), x - x^* \rangle \geq 0 \quad \text{for all } x \in \mathcal{X}, x^* \in \mathcal{X}^*. \tag{2.7}$$

By convexity, it follows that $\langle \nabla g(x), x - x^* \rangle < 0$ whenever $x^* \in \mathcal{X}^*$ and $x \in \mathcal{X} \setminus \mathcal{X}^*$, so equality holds in (2.7) if and only if $x \in \mathcal{X}^*$.

*Example* 2.2 (Pseudo/Quasi-convex programs). The previous example shows that variational coherence is a weaker and more general notion than convexity and/or operator monotonicity. In fact, as we show below, the class of variationally coherent problems also contains all *pseudo-convex* programs, i.e. when

$$\langle \nabla g(x), x' - x \rangle \geq 0 \implies g(x') \geq g(x), \tag{PC}$$

for all $x, x' \in \mathcal{X}$. In this case, we have:

**Proposition 2.4.** *If $g$ is pseudo-convex,* (Opt) *is variationally coherent.*

*Proof.* Take $x^* \in \mathcal{X}^*$ and $x \in \mathcal{X} \setminus \mathcal{X}^*$, and assume ad absurdum that $\langle \nabla g(x), x - x^* \rangle \leq 0$. By (PC), this implies that $g(x^*) \geq g(x)$, contradicting the choice of $x$ and $x^*$. We thus conclude that $\langle \nabla g(x), x - x^* \rangle > 0$ for all $x^* \in \mathcal{X}^*$, $x \in \mathcal{X} \setminus \mathcal{X}^*$; since $\langle \nabla g(x), x - x^* \rangle \leq 0$ if $x \in \mathcal{X}^*$, our claim follows by continuity. ∎

We recall that every convex function is pseudo-convex, and every pseudo-convex function is quasi-convex (i.e. its sublevel sets are convex). Both inclusions are proper, but the latter is fairly thin:

**Proposition 2.5.** *Suppose that $g$ is quasi-convex and non-degenerate, i.e.*

$$\langle g(x), z \rangle \neq 0 \quad \text{for all nonzero } z \in \mathrm{TC}(x), \ x \in \mathcal{X} \setminus \mathcal{X}^*, \tag{2.8}$$

*where $\mathrm{TC}(x)$ is the tangent cone vertexed at $x$. Then, $g$ is pseudo-convex (and variationally coherent).*

*Proof.* This follows from the following characterization of quasi-convex functions [6]: $g$ is quasi-convex if and only if $g(x') \leq g(x)$ implies that $\langle \nabla g(x), x' - x \rangle \leq 0$. By contraposition, this yields the strict part of (PC), i.e. $g(x') > g(x)$ whenever $\langle \nabla g(x), x' - x \rangle > 0$. To complete the proof, if $\langle \nabla g(x), x' - x \rangle = 0$ and $x \in \mathcal{X}^*$, (PC) is satisfied trivially; otherwise, if $\langle \nabla g(x), x' - x \rangle = 0$ but $x \in \mathcal{X} \setminus \mathcal{X}^*$, (2.8) implies that $x' - x = 0$, so $g(x') = g(x)$ and (PC) is satisfied as an equality. ∎

The non-degeneracy condition (2.8) is satisfied by every quasi-convex function after an arbitrarily small perturbation leaving its minimum set unchanged. By this token, Propositions 2.4 and 2.5 imply that essentially all quasi-convex programs are also variationally coherent.

*Example* 2.3 (Star-convex programs). If $g$ is star-convex, then $\langle \nabla g(x), x - x^* \rangle \geq g(x) - g(x^*)$ for all $x \in \mathcal{X}$, $x^* \in \mathcal{X}^*$. This is easily seen to be a special case of variational coherence because $\langle \nabla g(x), x - x^* \rangle \geq g(x) - g(x^*) \geq 0$, with the last inequality strict unless $x \in \mathcal{X}^*$. Note that star-convex functions contain convex functions as a subclass (but not necessarily pseudo/quasi-convex functions).

*Example* 2.4 (Beyond quasi-/star-convexity). A simple example of a function that is variationally coherent without being quasi-convex or star-convex is given by:

$$g(x) = 2 \sum_{i=1}^{d} \sqrt{1 + x_i}, \quad x \in [0, 1]^d. \tag{2.9}$$

When $d \geq 2$, it is easy to see $g$ is not quasi-convex: for instance, taking $d = 2$, $x = (0, 1)$ and $x' = (1, 0)$ yields $g(x/2 + x'/2) = 2\sqrt{6} > 2\sqrt{2} = \max\{g(x), g(x')\}$, so $g$ is not quasi-convex. It is also instantly clear this function is not star-convex even when $d = 1$ (in which case it is a concave function). On the other hand, to estabilish (VC), simply note that $\mathcal{X}^* = \{0\}$ and $\langle \nabla g(x), x - 0 \rangle = \sum_{i=1}^{d} x_i / \sqrt{1 + x_i} > 0$ for all $x \in [0, 1]^d \setminus \{0\}$. For a more elaborate example of a variationally coherent problem that is not quasi-convex, see Figure 2.

### 2.3 Stochastic Mirror Descent

To solve (Opt), we focus on the widely used family of algorithms known as stochastic mirror descent (SMD), formally given in Algorithm 1.[4] Heuristically, the main idea of the method is as follows: At each iteration, the algorithm takes as input an independent and identically distributed (i.i.d.) sample

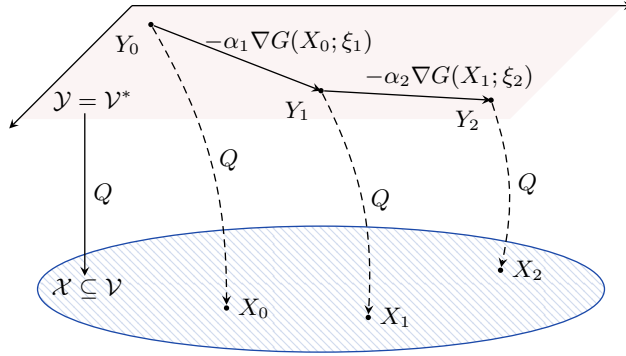

**Figure 1:** Schematic representation of stochastic mirror descent (Algorithm 1).

of the gradient of $G$ at the algorithm's current state. Subsequently, the method takes a step along this stochastic gradient in the dual space $\mathcal{Y} \equiv \mathcal{V}^*$ of $\mathcal{V}$ (where gradients live), the result is "mirrored" back to the problem's feasible region $\mathcal{X}$ to obtain a new solution candidate, and the process repeats.

In pseudocode form, we have:

---
**Algorithm 1** Stochastic mirror descent (SMD)

---
**Require:** Initial score variable $Y_0$
1: $n \leftarrow 0$
2: **repeat**
3:      $X_n = Q(Y_n)$
4:      $Y_{n+1} = Y_n - \alpha_{n+1} \nabla G(X_n, \xi_{n+1})$
5:      $n \leftarrow n + 1$
6: **until** end
7: **return** solution candidate $X_n$

---

In the above representation, the key elements of SMD (see also Fig. 1) are:

1. The "mirror map" $Q \colon \mathcal{Y} \to \mathcal{X}$ that outputs a solution candidate $X_n \in \mathcal{X}$ as a function of the auxiliary score variable $Y_n \in \mathcal{Y}$. In more detail, the algorithm's mirror map $Q$ is defined as

$$Q(y) = \arg\max_{x \in \mathcal{X}}\{\langle y, x \rangle - h(x)\}, \tag{2.10}$$

where $h(x)$ is a strongly convex function that plays the role of a regularizer. Different choices of the regularizer $h$ yields different specific algorithm. Due to space limitation, we mention in passing two well-known examples: When $h(x) = \frac{1}{2}\|x\|_2^2$ (i.e. Euclidean regularizer), mirror descent becomes gradient descent. When $h(x) = \sum_{i=1}^{d} x_i \log x_i$ (i.e. entropic regularizer), mirror descent becomes exponential gradient (aka exponential weights).

2. The step-size sequence $\alpha_n > 0$, chosen to satisfy the "$\ell^2 - \ell^1$" summability condition:

$$\sum_{n=1}^{\infty} \alpha_n^2 < \infty, \quad \sum_{n=1}^{\infty} \alpha_n = \infty. \tag{2.11}$$

3. A sequence of i.i.d. gradient samples $\nabla G(x; \xi_{n+1})$.[5]

## 3 Recurrence of SMD

In this section, we characterize an interesting recurrence phenomenon that will be useful later for establishing global convergence. Intuitively speaking, for a variationally coherent program of the

general form(Opt), any neighborhood of $\mathcal{X}^*$ will almost surely be visited by iterates $X_n$ infinitely often. Note that this already implies that at least a subsequence of iterates converges to global minima almost surely. To that end, we first define an important divergence measure between a primal variable $x$ and a dual variable $y$, called Fenchel coupling, that plays an indispensable role of an energy function.

**Definition 3.1.** Let $h\colon \mathcal{X} \to \mathbb{R}$ be a regularizer with respect to $\|\cdot\|$ that is $K$-strongly convex.

1. The convex conjugate function $h^* : \mathbb{R}^n \to \mathbb{R}$ of $h$ is defined as:

$$h^*(y) = \max_{x\in\mathcal{X}}\{\langle x, y\rangle - h(x)\}.$$

2. The *mirror map* $Q\colon \mathbb{R}^n \to \mathcal{X}$ associated with the regularizer $h$ is defined as:

$$Q(y) = \arg\max_{x\in\mathcal{X}}\{\langle x, y\rangle - h(x)\}.$$

3. The Fenchel coupling $F\colon \mathcal{X} \times \mathbb{R}^n \to \mathbb{R}$ is defined as:

$$F(x, y) = h(x) - \langle x, y\rangle + h^*(y).$$

Note that the naming of Fenchel coupling is natural as it consists of all the terms in the well-known Fenchel's inequality: $h(x) + h^*(y) \geq \langle x, y\rangle$. The Fenchel's inequality says that Fenchel coupling is always non-negative. As indicated by part 1 of the following lemma, a stronger result can be obtained. We state the two key properties Fenchel coupling next.

**Lemma 3.2.** *Let $h\colon \mathcal{X} \to \mathbb{R}$ be a $K$-strongly convex regularizer on $\mathcal{X}$. Then:*

1. $F(x, y) \geq \frac{1}{2}K\|Q(y) - x\|^2, \forall x \in \mathcal{X}, \forall y \in \mathbb{R}^n$.

2. $F(x, \tilde{y}) \leq F(x, y) + \langle \tilde{y} - y, Q(y) - x\rangle + \frac{1}{2K}\|\tilde{y} - y\|_*^2, \forall x \in \mathcal{X}, \forall \tilde{y}, y \in \mathbb{R}^n$.

We assume that we are working with mirror maps that are *regular* in the following weak sense:[6]

**Assumption 3.** The mirror map $Q$ is *regular*: if $Q(y_n) \to x$, then $F(x, y_n) \to 0$.

**Definition 3.3.** Given a point $x \in \mathcal{X}$, a set $\mathcal{S} \subset \mathcal{X}$ and a norm $\|\cdot\|$.

1. Define the point-to-set normed distance and Fenchel coupling distance respectively as: $\operatorname{dist}(x, \mathcal{S}) \triangleq \inf_{\mathbf{s}\in\mathcal{S}}\|x - \mathbf{s}\|$ and $F(\mathcal{S}, y) = \inf_{\mathbf{s}\in\mathcal{S}} F(\mathbf{s}, y)$.

2. Given $\varepsilon > 0$, define $B(\mathbf{S}, \varepsilon) \triangleq \{x \in \mathcal{X} \mid \operatorname{dist}(x, \mathcal{S}) < \varepsilon\}$.

3. Given $\delta > 0$, define $\tilde{B}(\mathcal{S}, \delta) \triangleq \{Q(y) \mid F(\mathcal{S}, y) < \delta\}$.

We then have the following recurrence result for a variationally coherent optimization problem Opt.

**Theorem 3.4.** *Under Assumptions 1–3, for any $\varepsilon > 0, \delta > 0$ and any $X_n$, the (random) iterates $\mathbf{X}_n$ generated in Algorithm 1 enter both $B(\mathcal{X}^*, \varepsilon)$ and $\tilde{B}(\mathcal{X}^*, \delta)$ infinitely often almost surely.*

## 4  Global Convergence Results

### 4.1  Deterministic Convergence

When a perfect gradient $\nabla g(x)$ is available (in Line 4 of Algorithm 1), SMD recovers its deterministic counterpart: mirror descent (Algorithm 2). We first characterize global convergence in this case.

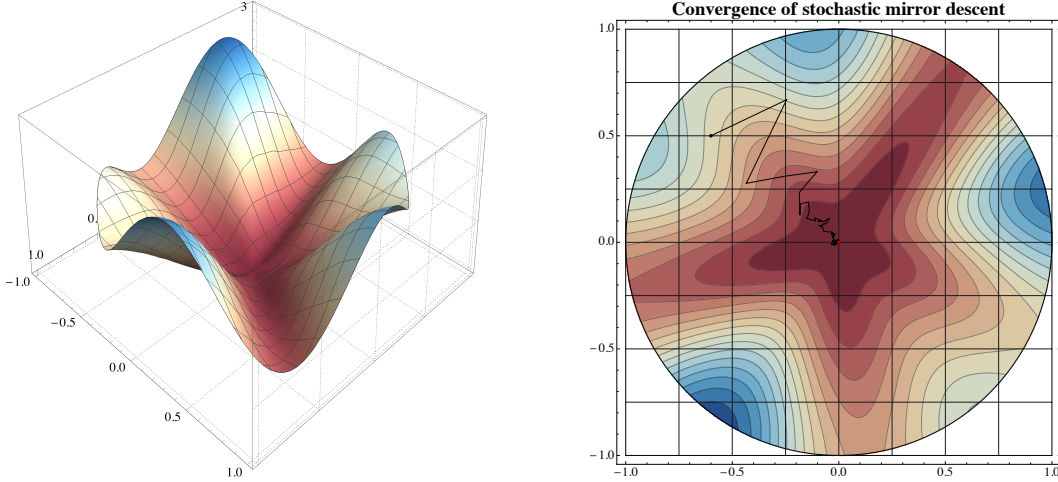

**Figure 2:** Convergence of stochastic mirror descent for the mean objective $g(r, \theta) = (2 + \cos\theta/2 + \cos(4\theta))r^2(5/3 - r)$ expressed in polar coordinates over the unit ball $(r \leq 1)$. In the left subfigure, we have plotted the graph of $g$; the plot to the right superimposes a typical SMD trajectory over the contours of $g$.

---

**Algorithm 2** Mirror descent (MD)

---

**Require:** Initial score variable $y_0$
1: $n \leftarrow 0$
2: **repeat**
3:      $x_n = Q(y_n)$
4:      $x_{n+1} = x_n - \alpha_{n+1}\nabla g(x_n)$
5:      $n \leftarrow n + 1$
6: **until** end
7: **return** solution candidate $x_n$

---

**Theorem 4.1.** *Consider an optimization problem Opt that is variationally coherent. Let $x_n$ be the iterates generated by MD. Under Assumption 3, $\lim_{t\to\infty} \text{dist}(x_n, \mathcal{X}^*) = 0$, for any $y_0$.*

*Remark* 4.1. Here we do not require $\nabla g(x)$ to be Lipschitz continuous. If $\nabla g(x)$ is indeed (locally) Lipschitz continuous, then Theorem 4.1 follows directly from Theorem 4.4. Otherwise, Theorem 4.1 requires a different argument, briefly outlined as follows. Theorem 3.4 implies that (in the special case of perfect gradient), iterates $x_n$ generated from MD enter $B(\mathcal{X}^*, \varepsilon)$ infinitely often. Now, by exploiting the properties of Fenchel coupling on a finer-grained level (compared to only using it to establish recurrence), we can establish that for any $\varepsilon$-neighborhood $B(\mathcal{X}^*, \varepsilon)$, after a certain number of iterations, once the iterate $x_n$ enters $B(\mathcal{X}^*, \varepsilon)$, it will never exit. Convergence therefore follows.

## 4.2 Stochastic Almost Sure Convergence

We begin with minimal mathematical preliminaries [4] needed that will be needed.

**Definition 4.2.** A semiflow $\Phi$ on a metric space $(M, d)$ is a continuous map $\Phi : \mathbb{R}_+ \times M \to M$:

$$(t, x) \to \Phi_t(x),$$

such that the semi-group properties hold: $\Phi_0$ = identity, $\Phi_{t+s} = \Phi_t \circ \Phi_s$ for all $(t, s) \in \mathbb{R}_+ \times \mathbb{R}_+$.

**Definition 4.3.** Let $\Phi$ be a semiflow on the metric space $(M, d)$. A continuous function $s : \mathbb{R}_+ \to M$ is an *asymptotic pseudotrajectory* (APT) for $\Phi$ if for every $T > 0$, the following holds:

$$\lim_{t\to\infty} \sup_{0 \leq h \leq T} d(s(t + h), \Phi_h(s(t))) = 0. \tag{4.1}$$

We are now ready to state the convergence result. See Figure 2 for a simulation example.

**Theorem 4.4.** *Consider an optimization problem Opt that is variationally coherent. Let $X_n$ be the iterates generated by SMD (Algorithm 1). Under Assumptions 1–3, if $\nabla g(x)$ is locally Lipschitz continuous on $\mathcal{X}$, then $\text{dist}(x_n, \mathcal{X}^*) \to 0$ almost surely as $t \to \infty$, irrespective of $Y_0$.*

*Remark* 4.2. The proof is rather involved and contains several ideas. To enhance the intuition and understanding, we outline the main steps here, each of which will be proved in detail in the appendix. To simplify the notation, we assume there is a unique optimal (i.e. $\mathcal{X}^*$ is a singleton set). The proof is identical in the multiple minima case, provide we replace $x^*$ by $\mathcal{X}^*$ and use the point-to-set distance.

1. We consider the following ODE approximation of SMD:

$$\dot{y} = v(x),$$
$$x = Q(y),$$

   where $v(x) = -\nabla g(x)$. We verify that the ODE admits a unique solution for $y(t)$ for any initial condition. Consequently, this solution induces a semiflow[7], which we denote $\Phi_t(y)$: it is the state at time $t$ given it starts at $y$ initially. Note that we have used $y$ as the initial point (as opposed to $y^0$) to indicate that the semiflow representing the solution trajectory should be viewed as a function of the initial point $y$.

2. We now relate the iterates generated by SMD to the above ODE's solution. Connect linearly the SMD iterates $Y_1, Y_2, \ldots, Y_k, \ldots$ at times $0, \alpha_1, \alpha_1 + \alpha_2, \ldots, \sum_{i=0}^{k-1} \alpha_i, \ldots$ respectively to form a continuous, piecewise affine (random) curve $Y(t)$. We then show that $Y(t)$ is almost surely an asymptotic pseudotrajectory of the semi-flow $\Phi$ induced by the above ODE.

3. Having characterized the relation between the SMD trajectory (affine interpolation of the discrete SMD iterates) and the ODE trajectory (the semi-flow), we now turn to studying the latter (the semiflow given by the ODE trajectory). A desirable property of $\Phi_t(y)$ is that the distance $F(x^*, \Phi_t(y))$ between the optimal solution $x^*$ and the dual variable $\Phi_t(y)$ (as measured by Fenchel coupling) can never increase as a function of $t$. We refer to this as the monotonicity property of Fenchel coupling under the ODE trajectory, to be contrasted to the discrete-time dynamics, where such monotonicity is absent (even when perfect information on the gradient is available). More formally, we show that $\forall y, \forall 0 \leq s \leq t$,

$$F(x^*, \Phi_s(y)) \geq F(x^*, \Phi_t(y)). \tag{4.2}$$

4. Continuing on the previous point, not only the distance $F(x^*, \Phi_t(y))$ can never increase as $t$ increases, but also, provided that $\Phi_t(y)$ is not too close to $x^*$, $F(x^*, \Phi_t(y))$ will decrease no slower than linearly. This suggests that either $\Phi_t(y)$ is already close to $x^*$ (and hence $x(t) = Q(\Phi_t(y))$ is close to $x^*$), or their distance will be decreased by a meaningful amount in (at least) the ensuing short time-frame. We formalize this discussion as follows:

$$\forall \varepsilon > 0, \forall y, \exists s > 0, F(x^*, \Phi_s(y)) \leq \max\{\frac{\varepsilon}{2}, F(x^*, y) - \frac{\varepsilon}{2}\}. \tag{4.3}$$

5. Now consider an arbitrary fixed horizon $T$. If at time $t$, $F(x^*, \Phi_0(Y(t)))$ is small, then by the monotonicity property in Claim 3, $F(x^*, \Phi_h(Y(t)))$ will remain small on the entire interval $h \in [0, T]$. Since $Y(t)$ is an asymptotic pseudotrajectory of $\Phi$ (Claim 2), $Y(t + h)$ and $\Phi_h(Y(t))$ should be very close for $h \in [0, T]$, at least for $t$ large enough. This means that $F(x^*, Y(t + h))$ should also be small on the entire interval $h \in [0, T]$. This can be made precise as follows: $\forall \varepsilon, T > 0, \exists \tau(\varepsilon, T) > 0$ such that $\forall t \geq \tau, \forall h \in [0, T]$:

$$F(x^*, Y(t + h)) < F(x^*, \Phi_h(Y(t))) + \frac{\varepsilon}{2}, \text{ a.s..} \tag{4.4}$$

6. Finally, we are ready to put the above pieces together. Claim 5 gives us a way to control the amount by which the two Fenchel coupling functions differ on the interval $[0, T]$. Claim 3 and Claim 4 together allow us to extend such control over successive intervals $[T, 2T), [2T, 3T), \ldots$, thereby establishing that, at least for $t$ large enough, if $F(x^*, Y(t))$ is small, then $F(x^*, Y(t + h))$ will remains small $\forall h > 0$. As it turns out, this means that after long enough time, if $x_n$ ever visits $\tilde{B}(x^*, \varepsilon)$, it will (almost surely) be forever trapped inside the neighborhood twice that size (i.e. $\tilde{B}(x^*, 2\varepsilon)$). Since Theorem 3.4 ensures that $x_n$ visits $\tilde{B}(x^*, \varepsilon)$ infinitively often (almost surely), the hypothesis is guaranteed to be true. Consequently, this leads to the following claim: $\forall \varepsilon > 0, \exists \tau_0$ (a positive integer), such that:

$$F(x^*, Y(\tau_0 + h)) < \varepsilon, \forall h \in [0, \infty), \text{ a.s..} \tag{4.5}$$

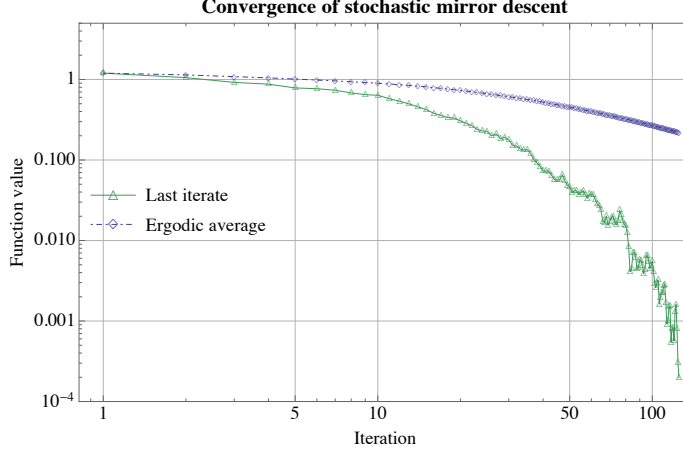

**Figure 3:** SMD run on the objective function of Fig. 2 with $\gamma_n \propto n^{-1/2}$ and Gaussian random noise with standard deviation about 150% the mean value of the gradient. Due to the lack of convexity, the algorithm's last iterate converges much faster than its ergodic average.

To conclude, Equation (4.5) implies that $F(x^*, Y_n) \to 0$, a.s. as $t \to \infty$, where the SMD iterates $Y_n$ are values at integer time points of the affine trajectory $Y(\tau)$. Per Statement 1 in Lemma 3.2, this gives $\|Q(Y_n) - x^*\| \to 0$, a.s. as $t \to \infty$, thereby establishing that $X_n = Q(Y_n) \to x^*$, a.s..

### 4.3 Convergence Rate Analysis

At the level of generality at which (VC) has been stated, it is unlikely that any convergence rate can be obtained, because unlike in the convex case, one has no handle on measuring the progress of mirror descent updates (recall that in (VC), only non-negativity is guaranteed for the inner product). Consequently, we focus here on the class of *strongly coherent* problems (a generalization of strongly convex problems) and derive a $\mathcal{O}(1/\sqrt{T})$ convergence rate in terms of the squared distance to a solution of (Opt).

**Definition 4.5.** We say that $g$ is *c-strongly variationally coherent* (or *c*-strongly coherent for short) if, for some $x^* \in \mathcal{X}$, we have:

$$\langle \nabla g(x), x - x^* \rangle \geq \frac{c}{2} \|x - x^*\|^2 \quad \text{for all } x \in \mathcal{X}. \tag{4.6}$$

**Theorem 4.6.** *If* (Opt) *is c-strongly coherent, then* $\|\bar{x}_T - x^*\|^2 \leq \frac{2}{c} \frac{F(x^*, y_0) + \frac{B}{2K} \sum_{n=0}^{T} \gamma_n^2}{\sum_{n=0}^{T} \gamma_n}$, *where* $\bar{x}_T = \frac{\sum_{n=0}^{T} \gamma_n x_n}{\sum_{n=0}^{T} \gamma_n}$, *K is the strong convexity coefficient of h and* $B = \max_{x \in \mathcal{X}} \| \nabla g(x)\|_*^2$.

The proof of Theorem 4.6 is given in the supplement. We mention a few implications of Theorem 4.6. First, in a strongly coherent optimization problem, if $\gamma_n = \frac{1}{\sqrt{n}}$, then $\|\bar{x}_T - x^*\|^2 = \mathcal{O}(\frac{\log T}{\sqrt{T}})$ (note that here $\ell^2 - \ell^1$ summability is not required for global convergence). By appropriately choosing the step-size sequence, one can further shave off the $\log T$ term above and obtain an $\mathcal{O}(1/\sqrt{T})$ convergence rate. This rate matches existing rates when applying gradient descent to strongly convex functions, although strongly variational coherence is a strict superset of strong convexity. Finally, note that even though we have characterized the rates in the mirror descent (i.e. perfect gradient case), one can easily obtain a mean $\mathcal{O}(1/\sqrt{T})$ rate in the stochastic case by using a similar argument. This discussion is omitted due to space limitation.

We end the section (and the paper) with an interesting observation from the simulation shown in Figure 3. The rate characterized in Theorem 4.6 is with respect to the ergodic average of the mirror descent iterates, while global convergence results established in Theorem 4.1 and Theorem 4.4 are both last iterate convergence. Figure 3 then provides a convergence speed comparison on the function given in Figure 2. It is apparent that the last iterate of SMD (more specifically, stochastic gradient descent) converges much faster than the ergodic average in this non-convex objective.

## 5 Acknowledgments

Zhengyuan Zhou is supported by Stanford Graduate Fellowship and would like to thank Yinyu Ye and Jose Blanchet for constructive discussions and feedback. Panayotis Mertikopoulos gratefully acknowledges financial support from the Huawei Innovation Research Program ULTRON and the ANR JCJC project ORACLESS (grant no. ANR–16–CE33–0004–01).

## Footnotes

[1]For the role of variational monotonicity in the context of convex programming, see also [22].

[2]For related approaches based on the theory of dynamical systems, see [21] and [12].

[3]In the above, gradients are treated as elements of the dual space $\mathcal{V}^*$ of $\mathcal{V}$ and $\|v\|_* = \sup\{\langle v, x \rangle : \|x\| \leq 1\}$ denotes the dual norm of $v \in \mathcal{V}^*$. We also note that $\nabla G(x;\xi)$ refers to the gradient of $G(x;\xi)$ with respect to $x$; since $\Xi$ need not have a differential structure, there is no danger of confusion.

[4] Mirror descent dates back to the original work of Nemirovski and Yudin [16]. More recent treatments include [1, 8, 15, 18, 20] and many others; the specific variant of SMD that we are considering here is most closely related to Nesterov's "dual averaging" scheme [18].

[5]The specific indexing convention for $\xi_n$ has been chosen so that $Y_n$ and $X_n$ are both adapted to the natural filtration $\mathcal{F}_n$ of $\xi_n$.

[6]Mirror maps induced by many common regularizers are regular, including the Euclidean regularizer and the entropic regularizer.

[7]A crucial point to note is that since $C$ may not be invertible, there may not exist a unique solution for $x(t)$.

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
