[Supplementary Material]

# Stochastic Mirror Descent in Variationally Coherent Optimization Problems Supplementary Material

**Zhengyuan Zhou**
Stanford University
zyzhou@stanford.edu

**Panayotis Mertikopoulos**
Univ. Grenoble Alpes, CNRS, Inria, LIG
panayotis.mertikopoulos@imag.fr

**Nicholas Bambos**
Stanford University
bambos@stanford.edu

**Stephen Boyd**
Stanford University
boyd@stanford.edu

**Peter Glynn**
Stanford University
glynn@stanford.edu

## Abstract

This appendix provides the missing proofs in the paper. Except for the first section, all sections match those of the main paper. Since the numberings are different, the results from the main paper are stated again before the proofs are given.

## 1 Martingale Convergence Theorems

We start by stating two Martingale convergence theorems that shall be useful later. These are adapted statements that come from [3], which contains detailed proofs. The first one is a law of large number theorem for Martingales.

**Theorem 1.1.** *Let $S^t = \sum_{k=0}^{t} X^k$ be a Martingale adapted to the filtration $\mathcal{S}^t$. Let $\{u^t\}_{t=0}^{\infty}$ be a nondecreasing sequence of positive numbers with $\lim_{t\to\infty} u^t = \infty$. If $\exists p \in [1,2]$ such that $\sum_{t=0}^{\infty} \frac{\mathbf{E}[|X^{t+1}|^p|\mathcal{S}^t]}{(u^t)^p} < \infty$, a.s. , then*

$$\lim_{t\to\infty} \frac{S^t}{u^t} = 0, \ a.s.,$$

The second one is Doob's Martingale convergence theorem.

**Theorem 1.2.** *Let $S^t$ be a submartingale adapted to the filtration $\mathcal{S}^t$, where $t = 0, 1, 2, \ldots$. If $S^t$ is $l^1$-bounded: $\sup_{t \geq 0} \mathbf{E}[|S^t|] < \infty$, then $S^t$ converges almost surely to a random variable $S$ with $\mathbf{E}[|S|] < \infty$.*

## 2 Setup and Preliminaries

### 2.1 Variational Coherence

**Lemma 2.1.** *Given $g \colon \mathcal{X} \to \mathbb{R}$, if $\mathcal{X}^*$ is a non-empty variationally coherent set, then $\mathcal{X}^*$ is a closed and convex set of all the global minima of $g$.*

*Proof.* First we show that any $\mathbf{x}^* \in \mathcal{X}^*$ is a global minimum. Pick any $x \in \mathcal{X}$ and denote $z = x - x^*$. Let $\tau \in [0,1]$. We have:

$$\frac{df(x^* + \tau z)}{d\tau} = \langle \nabla f(x^* + \tau z), z \rangle = \frac{1}{\tau} \langle \nabla f(x^* + \tau z), x^* + \tau z - x^* \rangle \geq 0, \qquad (2.1)$$

where the last inequality follows from variational stability. Consequently, $f(x^* + \tau z)$ is increasing in $\tau$, thereby implying (by setting $\tau = 1$) that $f(x^*) \leq f(x^* + z) = f(x)$. Since $x$ is arbitrary, $\mathbf{x}^*$ must be a global minimum.

Next, we show that $\mathcal{X}^*$ is closed. Take any converging sequence $\{\mathbf{x}^j\}_{j=0}^\infty$ in $\mathcal{X}^*$ with $\lim_{j\to\infty} \mathbf{x}^j = \mathbf{x}^*$. Then, for any $\mathbf{x} \in \mathcal{X}$, since $x^j \in \mathcal{X}^*$, we have

$$\langle \nabla f(\mathbf{x}), x - x^j \rangle \geq 0, \forall j = 0, 1, \dots$$

It follows that

$$\lim_{j\to\infty} \langle \nabla f(\mathbf{x}), x - x^j \rangle = \langle \nabla f(\mathbf{x}), x - x^* \rangle \geq 0, \forall \mathbf{x} \in \mathcal{X},$$

thereby implying $\mathbf{x}^* \in \mathcal{X}^*$. Since $\{\mathbf{x}^j\}_{j=0}^\infty$ is any sequence in $\mathcal{X}^*$, $\mathcal{X}^*$ contains all its limit points and is therefore closed.

To see that $\mathcal{X}^*$ is convex, take any $\mathbf{x}^*, \mathbf{z}^* \in \mathcal{X}^*$ and any $\tau \in [0, 1]$. For any $\mathbf{x} \in \mathcal{X}$, we have

$$\langle \nabla f(\mathbf{x}), x - (\tau x^* + (1-\tau) z^*) \rangle = \tag{2.2}$$
$$\tau \langle \nabla f(\mathbf{x}), x - x^* \rangle + (1-\tau)\langle \nabla f(\mathbf{x}), x - z^* \rangle \geq 0, \tag{2.3}$$

thereby establishing that $\tau \mathbf{x}^* + (1-\tau)\mathbf{z}^* \in \mathcal{X}^*$.

Finally, to see that $\mathcal{X}^*$ contains all global minima, assume that $\mathbf{z}^* \notin \mathcal{X}^*$ is a global minimum. Per Equation 2.1, and per the definition of variational stability, we have

$$\frac{df(x^* + \tau z)}{d\tau} = \langle \nabla f(x^* + \tau z), z \rangle = \frac{1}{\tau}\langle \nabla f(x^* + \tau z), x^* + \tau z - x^* \rangle > 0,$$

when $x^* + \tau z \notin \mathcal{X}^*$. By setting $\tau = 1, z = z^* - x^*$, we then have $f(z^*) > f(x^*)$, a contradiction that $z^*$ is a minimum. □

## 2.2 Examples of Variational Coherence

As an example optimization problem that is nearly convex but not quasi-convex, fix any $d > 1$, consider:

$$\min_{\mathbf{x}\in[0,1]^d} \sum_{i=1}^d \sqrt{1 + x_i}.$$

It can be easily checked that $g(\mathbf{x}) = \sum_{i=1}^d \sqrt{1 + x_i}$ is not quasi-convex: for example, take $d = 2$, $\mathbf{x} = (0, 1), \mathbf{z} = (1, 0)$, we have $g(0.5\mathbf{x} + 0.5\mathbf{z}) > \max\{g(\mathbf{x}), g(\mathbf{z})\}$. On the other hand, to see it is nearly convex, take $\mathbf{X}^* = \{\mathbf{0}\}$. It follows that $\langle \nabla f(\mathbf{x}), \mathbf{x} - \mathbf{0} \rangle = \sum_{i=1}^d \frac{x_i}{2\sqrt{1+x_i}} > 0, \forall \mathbf{x} \in [0,1]^d - \{\mathbf{0}\}$,

# 3 Recurrence of SMD

**Lemma 3.1.** *Let $h\colon \mathcal{X} \to \mathbb{R}$ be a regularizer with respect to $\|\cdot\|$ that is $K$-strongly convex. Then:*

1. $F(x, y) \geq \frac{1}{2}K\|C(y) - x\|^2, \forall x \in \mathcal{X}, \forall y \in \mathbb{R}^n$.

2. $F(x, \tilde{y}) \leq F(x, y) + \langle \tilde{y} - y, C(y) - x \rangle + \frac{1}{2K}\|\tilde{y} - y\|_*^2, \forall x \in \mathcal{X}, \forall \tilde{y}, y \in \mathbb{R}^n$.

*Proof.* To prove the first claim, let $p = C(y)$. It follows from standard convex analysis [5] that $y \in \partial h(p)$ ($y$ is a subgradient of $h(p)$ if and only if $p = C(y)$). By definition:

$$F(x, y) = h(x) + h^*(y) - \langle y, x \rangle == h(x) + \langle y, C(y) \rangle - h(C(y)) - \langle y, x \rangle = h(x) - h(p) - \langle y, x - p \rangle. \tag{3.1}$$

Since $y \in \partial h(p)$, and $h$ is $K$-strongly convex, we have that $\forall t \in (0, 1]$:

$$h(p) + t\langle y, x - p \rangle \leq h(p + t(x - p))$$
$$\leq th(x) + (1-t)h(p) - \frac{1}{2}Kt(1-t)\|p - x\|^2, \tag{3.2}$$

thereby leading to the bound

$$\tfrac{1}{2}K(1-t)\|p-x\|^2 \le h(x) - h(p) - \langle y, x-p \rangle = F(x,y), \tag{3.3}$$

for all $t \in (0,1]$. The claim then follows by letting $t \to 0^+$ in (3.3).

For the second claim, we start by citing here a useful result in [6] (Theorem 12.60): For a proper, lower semi-continuous and convex function $f : \mathbb{R}^n \to \bar{\mathbb{R}}$, where $\bar{\mathbb{R}} = [-\infty, \infty]$ and a value $\sigma > 0$, $f^*$ is $\sigma$-strongly convex (with respect to norm $\|\cdot\|_*$) if and only if $f$ is differentiable and satisfies:

$$f(\tilde{x}) \le f(x) + \langle \nabla f(x), \tilde{x} - x \rangle + \frac{1}{2\sigma}\|\tilde{x} - x\|^2, \forall x, \tilde{x}.$$

Next, we note that in our case, $h$ is $K$-strongly convex with respect to norm $\|\cdot\|_i$ and since $h$ is proper, lower semi-continuous and convex, it follows that $(h^*)^* = h$ (Theorem 11.1 in [6]). Further, it can be easily checked that $h^*$ is proper, lower semi-continuous and convex (since it is a point-wise maximum of affine functions per its definition), it therefore follows that the $K$-strong convexity of $(h^*)^*$ (with respect to $\|\cdot\|_{**} = \|\cdot\|$) implies that $h^*$ is differentiable and satisfies:

$$h^*(\tilde{y}) \le h^*(y) + \langle \tilde{y} - y, \nabla h^*(y) \rangle + \frac{1}{2K}\|\tilde{y} - y\|_*^2, \forall y, \tilde{y} \tag{3.4}$$

$$= h^*(y) + \langle \tilde{y} - y, C(y) \rangle + \frac{1}{2K}\|\tilde{y} - y\|_*^2, \forall y, \tilde{y} \tag{3.5}$$

where the equality follows because $\nabla h^*(y) = C(y)$.

Therefore, upon substituting the preceding inequality into the definition of Fenchel coupling, we have:

$$F(x, \tilde{y}) = h(x) + h^*(\tilde{y}) - \langle \tilde{y}, x \rangle \tag{3.6}$$

$$\le h(x) + h^*(y) + \langle \tilde{y} - y, \nabla h^*(y) \rangle + \frac{1}{2K}\|\tilde{y} - y\|_*^2 - \langle \tilde{y}, x \rangle \tag{3.7}$$

$$= F(x, y) + \langle \tilde{y} - y, C(y) - x \rangle + \frac{1}{2K}\|\tilde{y} - y\|_*^2, \tag{3.8}$$

thereby establishing the claim. □

**Assumption 1.** $G(x, \xi)$ is continuously differentiable in $x$ for almost all $\xi \in \Omega$.

**Assumption 2.** $\nabla G(x; \xi)$ has bounded second moments and is Lipschitz continuous in the mean: $\mathbb{E}[\|\nabla G(x; \xi)\|_*^2] < \infty$ for all $x \in \mathcal{X}$ and $\mathbb{E}[\nabla G(x; \xi)]$ is Lipschitz on $\mathcal{X}$.[1]

**Assumption 3.** The mirror map $C$ is *regular*: if $C(y_t) \to x$, then $F(x, y_t) \to 0$.

**Theorem 3.2.** *Under Assumptions 1–3, for any $\varepsilon > 0, \delta > 0$ and any $x_t$, the (random) iterates $X_t$ generated in Algorithm 1 enter both $B(\mathcal{X}^*, \varepsilon)$ and $\tilde{B}(\mathcal{X}^*, \delta)$ infinitely often almost surely.*

*Proof.* We break the proof into 3 steps.

1. Denote $v(x) = -\mathbb{E}_{\xi_{t+1} \sim \Pi}[\nabla G(x, \xi_{t+1})]$. Per Assumption 2, we have $\mathbb{E}_{\xi_{t+1} \sim \Pi}[\|\nabla G(x, \xi_{t+1})\|_*] < \infty$, since finite second moments directly imply finite first moments. This implies:

$$v(x) = -\nabla \mathbb{E}_{\xi_{t+1} \sim \Pi}[G(x, \xi_{t+1})] = -g(x),$$

where $v(x)$ is also continuous on $\mathcal{X}$ per Assumption 1. Denote also $\tilde{v}_t(x) = -\nabla G(x, \xi_{t+1})$, we have $\mathbb{E}[\|\tilde{v}_t(x)\|_*] < \infty$ and $\mathbb{E}[\|\tilde{v}_t(x)\|_*]$ is a continuous function on $\mathcal{X}$. The gradient update in Algorithm 1 can be rewritten as:

$$Y_{t+1} = Y_t + \alpha_t \left\{ v(X_t) + \tilde{v}_t(X_t) - v(X_t) \right\}. \tag{3.9}$$

Note that the first term $v(X_t)$ is the "ideal" gradient and the second term $\tilde{v}_t(X_t) - v(X_t)$ is a martingale difference adapted to $\xi_0, \ldots, \xi_t$ because:

(a) Under the above notation and bounded first moment conclusions, it follows:

$$\mathbb{E}[\|\tilde{v}_t(X_t) - v(X_t)\|_*] \leq \mathbb{E}[\|\tilde{v}_t(X_t)\|_*] + \mathbb{E}[\|v(X_t)\|_*] \tag{3.10}$$

$$\leq \sup_{x \in \mathcal{X}} \mathbb{E}[\|\tilde{v}_t(x)\|_*] + \sup_{x \in \mathcal{X}} \|v(x)\|_* < \infty, \tag{3.11}$$

where the last inequality follows from $\mathcal{X}$ is compact.

(b) The martingale difference property holds:

$$\mathbb{E}[\tilde{v}_t(X_t) - v(X_t) \mid \xi_0, \dots, \xi_t] = \mathbb{E}[\tilde{v}_t(X_t) \mid \xi_0, \dots, \xi_t] - v(X_t) = v(X_t) - v_t(X_t) = 0,$$

where the first equality follows from the fact that $X_t$ is adapted to $\xi_0, \dots, \xi_t$ and the second equality follows from the definition of $\tilde{v}_t$ and the independence of $\xi_t$'s.

Furthermore, this martingale has uniformly bounded conditional second moments:

$$\forall t, \mathbb{E}[\|\tilde{v}_t(X_t) - v(X_t)\|_*^2 \mid \xi_0, \dots, \xi_t] \leq V_* < \infty, \text{a.s.}, \tag{3.12}$$

for some $V_*$. This can be seen by noting that per Assumption 2, $\mathbb{E}[\|\tilde{v}_t(x)\|_*^2] = \mathbb{E}_{\xi_{t+1} \sim \Pi}[\|\nabla G(x, \xi_{t+1})\|_*^2] < \infty$ and per Assumption 1, $\mathbb{E}[\|\tilde{v}_t(x)\|_*^2]$ is a continuous function on the compact set $\mathcal{X}$ and therefore bounded:

$$B \triangleq \sup_{x \in \mathcal{X}} \mathbb{E}[\|\tilde{v}_t(x)\|_*^2]. \tag{3.13}$$

Further, by Jensen's inequality, we have

$$\sup_{x \in \mathcal{X}} \|v(x)\|_*^2 = \sup_{x \in \mathcal{X}} \|\mathbb{E}[\nabla G(x, \xi_{t+1})]\|_*^2 \leq \sup_{x \in \mathcal{X}} \mathbb{E}[\|\nabla G(x, \xi_{t+1})\|_*^2] = B, \tag{3.14}$$

where the last inequality follows from Equation 3.13. Consequently, for every $t$:

$$\mathbb{E}[\|\tilde{v}_t(X_t) - v(X_t)\|_*^2 \mid \xi_0, \dots, \xi_t] \leq 2\left\{\mathbb{E}[\|\tilde{v}_t(X_t)\|_*^2 \mid \xi_0, \dots, \xi_t] + \mathbb{E}[\|v(X_t)\|_*^2 \mid \xi_0, \dots, \xi_t]\right\} \tag{3.15}$$

$$= 2\left\{\mathbb{E}[\|\tilde{v}_t(X_t)\|_*^2 \mid \xi_0, \dots, \xi_t] + \|v(X_t)\|_*^2\right\} \leq 2(B + B) = 4B \stackrel{\triangle}{=} V_*, \tag{3.16}$$

thereby establishing Equation 3.12.

2. Per the previous claim, denote the martingale difference by $\tilde{\xi}_{t+1} \triangleq \tilde{v}_t(X_t) - v_t(X_t)$. We can rewrite Equation (4.6) as:

$$Y_{t+1} = Y_t + \alpha_t \left\{v(X_t) + \tilde{\xi}_{t+1}\right\}. \tag{3.17}$$

Using Fenchel coupling as a Lyapunov function and various martingale convergence results, we establish that unless $X_t$ enters $B(\mathcal{X}^*, \varepsilon)$ infinitely often, the Fenchel coupling values will diverge to $-\infty$, in which case it contradicts the fact that it is always non-negative (first claim in Lemma 3.1).

Let $Y_t, X_t$ be the iterates generated in Algorithm 1. Fix an arbitrary $\varepsilon > 0$. Assume for contradiction purposes that $X_t$ enters visits $B(\mathcal{X}^*, \varepsilon)$ a finite number of times and hence let $t^0 - 1$ be the last time $X_t$ is in $B(\mathcal{X}^*, \varepsilon)$: $\forall t \geq t^0, X_t \in \mathcal{X} - B(\mathcal{X}^*, \varepsilon)$. Since $\mathcal{X} - B(\mathcal{X}^*, \varepsilon)$ is a compact set and $v(x) = -\nabla g(x)$ is continuous in $x$ (from the previous step) and since $\langle \nabla v(x), x - x^* \rangle = -\langle \nabla g(x), x - x^* \rangle < 0, \forall x \in \mathcal{X}, x \notin \mathcal{X}^*$ by variational coherence, it follows that there exists a $c_{\max}(\varepsilon) < 0$ such that

$$\langle \nabla v(x), x - x^* \rangle \leq c_{\max}(\varepsilon), \forall x \in \mathcal{X} - B(\mathcal{X}^*, \varepsilon). \tag{3.18}$$

Next denote $R = \max_{x \in \mathcal{X}} \|x\|$ and $\beta_t \triangleq \frac{\alpha_t^2}{2K}$ and note that $\sum_{t=1}^{\infty} \beta_t < \infty$. Following the notation in the previous step, $\tilde{B} \triangleq \max_{x \in \mathcal{X}} \|v(x)\|_*^2 < \infty$. Using Lemma 3.1, we have $\forall t \geq t^0$ and $\forall x^* \in \mathcal{X}^*$:

$$F(x^*, Y_{t+1}) = F(x^*, Y_t + \alpha_t(v(X_t) + \tilde{\xi}_{t+1})) \tag{3.19}$$

$$\leq F(x^*, Y_t) + \alpha_t \langle v(X_t) + \tilde{\xi}_{t+1}, C(Y_t) - x^* \rangle + \beta_t \|v(X_t) + \tilde{\xi}_{t+1}\|_*^2 \tag{3.20}$$

$$= F(x^*, Y_t) + \alpha_t \langle v(X_t), X_t - x^* \rangle + \alpha_t \langle \tilde{\xi}_{t+1}, X_t - x^* \rangle + \beta_t \|v(X_t) + \tilde{\xi}_{t+1}\|_*^2 \tag{3.21}$$

$$\leq F(x^*, Y_t) + \alpha_t c_{\max}(\varepsilon) + \alpha_t \langle \tilde{\xi}_{t+1}, X_t - x^* \rangle + 2\beta_t \left\{ \|v(X_t)\|_* + \|\tilde{\xi}_{t+1}\|_*^2 \right\} \tag{3.22}$$

$$\leq F(x^*, Y_{t^0}) + (\sum_{k=t^0}^{t} \alpha_k) c_{\max}(\varepsilon) + \sum_{k=t^0}^{t} \alpha_k \langle \tilde{\xi}_{k+1}, X_k - x^* \rangle + 2 \sum_{k=t^0}^{t} \beta_k \left\{ \|v(X_k)\|_*^2 + \|\tilde{\xi}_{k+1}\|_*^2 \right\} \tag{3.23}$$

$$= F(x^*, Y_{t^0}) + (\sum_{k=t^0}^{t} \alpha_k) \left\{ c_{\max}(\varepsilon) + \sum_{k=t^0}^{t} \frac{\alpha_k}{\sum_{k=t^0}^{t} \alpha_k} \langle \tilde{\xi}_{k+1}, X_k - x^* \rangle \right\} \tag{3.24}$$

$$+ 2 \sum_{k=t^0}^{t} \beta_k \left\{ \|v(X_k)\|_*^2 + \|\tilde{\xi}_{k+1}\|_*^2 \right\} \tag{3.25}$$

$$= F(x^*, Y_{t^0}) + (\sum_{k=t^0}^{t} \alpha_k) \left\{ c_{\max}(\varepsilon) + \sum_{k=t^0}^{t} \frac{\alpha_k}{\sum_{k=t^0}^{t} \alpha_k} \langle \tilde{\xi}_{k+1}, X_k - x^* \rangle \right\} \tag{3.26}$$

$$+ 2 \sum_{k=t^0}^{t} \beta_k \|\tilde{\xi}_{k+1}\|_*^2 + 2 \sum_{k=t^0}^{t} \beta_k \tilde{B}. \tag{3.27}$$

Next, note that $\sum_{k=t^0}^{t} \alpha^k \langle \tilde{\xi}_{k+1}, (X_k - x^*) \rangle$ is a martingale adapted to $\mathcal{F}^{t+1} = \sigma(\tilde{\xi}_1, \ldots, \tilde{\xi}_{t+1})$ (the filtration generated by the random variables $\tilde{\xi}_1, \ldots, \tilde{\xi}_{t+1}$) because

$$\mathbf{E}[\alpha_k \langle \tilde{\xi}_{t+1}, X_t - x^* \rangle | \mathcal{F}^t] = \alpha_k \langle \mathbf{E}[\tilde{\xi}_{t+1} | \mathcal{F}^t], X_t - x^* \rangle = 0,$$

where the first equality follows since $X_t$ is adapted to $\mathcal{F}^t$. Furthermore, setting $p = 2$ and $u^t = \sum_{k=t^0}^{t} \alpha_k$ (the first $t^0$ terms of $u^t$ can be set arbitrarily and are not essential), it is clear that $u^t$ is increasing and $\lim_{t \to \infty} u^t = \infty$. It then follows that

$$\sum_{t=0}^{\infty} \frac{\mathbf{E}[|\alpha_t \tilde{\xi}_{t+1}|^p \mid \mathcal{F}^t]}{(u^t)^p} \leq \sum_{t=0}^{\infty} \frac{(\alpha_t)^2 V_*}{(u^t)^2} \leq V_* \sum_{t=0}^{\infty} \frac{(\alpha_t)^2}{(\sum_{k=t^0}^{t} \alpha_k)^2} < \infty, \text{ a.s.},$$

where the last inequality follows from the fact that $\alpha_t$ is square-summable. Consequently, by Theorem 1.1,

$$\sum_{k=t^0}^{t} \frac{\alpha_k}{\sum_{k=t^0}^{t} \alpha_k} \langle \tilde{\xi}_{k+1}, X_k - x^* \rangle \to 0, \text{ a.s., } t \to \infty.$$

Since $\sum_{k=t^0}^{\infty} \alpha_k = -\infty$, we therefore have

$$(\sum_{k=t^0}^{t} \alpha_k) \left\{ c_{\max}(\epsilon) + \sum_{k=t^0}^{t} \frac{\alpha_k}{\sum_{k=t^0}^{t} \alpha_k} \langle \tilde{\xi}_{k+1}, X_k - x^* \rangle \right\} \to -\infty, \text{ a.s., } t \to \infty.$$

Finally, note that since $\| \cdot \|_*$ is a norm and hence convex, $S^t = \sum_{k=t^0}^{t} \beta_k \|\tilde{\xi}_{k+1}\|_*^2$ is a submartingale adapted to $\mathcal{F}^{t+1}$. We check that $S^t$ is $l_1$-bounded:

$$\mathbf{E}[S^t] = \sum_{k=t^0}^{t} \beta_k \mathbf{E}[\|\tilde{\xi}_{k+1}\|_*^2] \leq \sum_{k=t^0}^{t} \beta_k V_* < V_* \sum_{k=t^0}^{\infty} \beta_k < \infty, \forall t,$$

where the first inequality follows from Equation 3.12.

Consequently, by Theorem 1.2, it follows that $S^t \to S$, a.s., $t \to \infty$, with $S$ finite almost surely. This then leads to the following statement (which holds almost surely):

$$(\sum_{k=t^0}^{t} \alpha_k) \left\{ c_{\max}(\varepsilon) + \sum_{k=t^0}^{t} \frac{\alpha_k}{\sum_{k=t^0}^{t} \alpha_k} \langle \tilde{\xi}_{k+1}, X_k - x^* \rangle \right\} + 2 \sum_{k=t^0}^{t} \beta_k \|\tilde{\xi}_{k+1}\|_*^2 \to -\infty.$$

Finally, Equation (3.26) then implies that $F(x^*, y_t) \to -\infty$, which contradicts the first statement in Lemma 3.1. The claim therefore follows.

3. Using the regularity of the mirror map, we show that $\tilde{B}(\mathcal{X}^*, \delta)$ always contains an open ball within it:

$$\forall \delta > 0, \exists \varepsilon(\delta) > 0, B(\mathcal{X}^*, \varepsilon) \subset \tilde{B}(\mathcal{X}^*, \delta). \tag{3.28}$$

Since $X_t$ enters $B(\mathcal{X}^*, \varepsilon)$ infinitely often almost surely, it must therefore enter $\tilde{B}(\mathcal{X}^*, \delta)$ infinitely often, almost surely.

Assume for contradiction purposes no $B(\mathcal{X}^*, \varepsilon)$ is contained in $\tilde{B}(\mathcal{X}^*, \delta)$, which means that for any $\delta > 0, \exists y^l$, such that $\text{dist}(Q(y_l), -\mathcal{X}^*) = \delta$ but $F^\lambda(\mathcal{X}^*, y_l) \geq \varepsilon$. This produces a sequence $\{y_l\}_{l=0}^\infty$ such that $C(y_l) \to \mathcal{X}^*$ but $F^\lambda(\mathcal{X}^*, y_l) \geq \varepsilon, \forall l$. This contradicts with the fact that the choice map $C(\cdot)$ is $\lambda$-Fenchel coupling conforming: by definition it holds that if $C(y_t) \to x$, then $F^\lambda(x, y_t) \to 0$ and since $\mathcal{X}^*$ is closed, we have $C(y_t) \to \mathcal{X}^*$ implies $F^\lambda(\mathcal{X}^*, y_t) \to 0$. Consequently, the claim follows.

$\square$

# 4 Global Convergence Results

We consider here again the problem:

$$\begin{aligned}
\text{minimize} \quad & g(x), \\
\text{subject to} \quad & x \in \mathcal{X},
\end{aligned} \tag{Opt}$$

We seek to prove the following:

**Theorem 4.1.** *Consider an optimization problem Opt that is variationally coherent. Let $x_t$ be the iterates generated by SMD (Algorithm 1). Under Assumptions 1–3, if $\nabla g(x)$ is locally Lipschitz continuous on $\mathcal{X}$, then $\text{dist}(x_t, \mathcal{X}^*) \to 0$ almost surely as $t \to \infty$, irrespective of $Y_0$.*

*Remark* 4.1. We repeat the 6 main ingredients here again.

1. We consider the following ordinary differential equation (ODE) approximation of SMD:

$$\begin{aligned}
\dot{y} &= v(x), \\
x &= C(y),
\end{aligned}$$

where following the previous notation $v(x) = -\nabla g(x)$. We verify that the ODE admits a unique solution for $y(t)$ for any initial condition. Consequently, this solution induces a semiflow[2], which we denote $\Phi_t(y)$: it is the state at time $t$ given it starts at $y$ initially. Note that we have used $y$ as the initial point (as opposed to $y^0$) to indicate that the semiflow representing the solution trajectory should be viewed as a function of the initial point $y$.

2. We now relate the iterates generated by SMD to the above ODE's solution. Connect linearly the SMD iterates $Y_1, Y_2, \ldots, Y_k, \ldots$ at times $0, \alpha_1, \alpha_1 + \alpha_2, \ldots, \sum_{i=0}^{k-1} \alpha_i, \ldots$ respectively to form a continuous, piecewise affine (random) curve $Y(t)$. We then show that $Y(t)$ is almost surely an asymptotic pseudotrajectory of the semi-flow $\Phi$ induced by the above ODE.

3. Having characterized the relation between the SMD trajectory (affine interpolation of the discrete SMD iterates) and the ODE trajectory (the semi-flow), we now turn to studying the latter (the semiflow given by the ODE trajectory). A desirable property of $\Phi_t(y)$ is that the distance $F(x^*, \Phi_t(y))$ between the optimal solution $x^*$ and the dual variable $\Phi_t(y)$ (as measured by Fenchel coupling) can never increase as a function of $t$. We refer to this as the monotonicity property of Fenchel coupling under the ODE trajectory, to be contrasted to the discrete-time dynamics, where such monotonicity is absent (even when perfect information on the gradient is available). More formally, we show that $\forall y, \forall 0 \leq s \leq t$,

$$F(x^*, \Phi_s(y)) \geq F(x^*, \Phi_t(y)). \tag{4.1}$$

4. Continuing on the previous point, not only the distance $F(x^*, \Phi_t(y))$ can never increase as $t$ increases, but also, provided that $\Phi_t(y)$ is not too close to $x^*$, $F(x^*, \Phi_t(y))$ will decrease no slower than linearly. This suggests that either $\Phi_t(y)$ is already close to $x^*$ (and hence $x(t) = C(\Phi_t(y))$ is close to $x^*$), or their distance will be decreased by a meaningful amount in (at least) the ensuing short time-frame. We formalize this discussion as follows:

$$\forall \varepsilon > 0, \forall y, \exists s > 0, F(x^*, \Phi_s(y)) \leq \max\{\frac{\varepsilon}{2}, F(x^*, y) - \frac{\varepsilon}{2}\}. \tag{4.2}$$

5. Now consider an arbitrary fixed horizon $T$. If at time $t$, $F(x^*, \Phi_0(Y(t)))$ is small, then by the monotonicity property in Claim 3, $F(x^*, \Phi_h(Y(t)))$ will remain small on the entire interval $h \in [0, T]$. Since $Y(t)$ is an asymptotic pseudotrajectory of $\Phi$ (Claim 2), $Y(t + h)$ and $\Phi_h(Y(t))$ should be very close for $h \in [0, T]$, at least for $t$ large enough. This means that $F(x^*, Y(t + h))$ should also be small on the entire interval $h \in [0, T]$. This can be made precise as follows: $\forall \varepsilon, T > 0, \exists \tau(\varepsilon, T) > 0$ such that $\forall t \geq \tau, \forall h \in [0, T]$:

$$F(x^*, Y(t + h)) < F(x^*, \Phi_h(Y(t))) + \frac{\varepsilon}{2}, \quad \text{a.s.}. \tag{4.3}$$

6. Finally, we are ready to put the above pieces together. Claim 5 gives us a way to control the amount by which the two Fenchel coupling functions differ on the interval $[0, T]$. Claim 3 and Claim 4 together allow us to extend such control over successive intervals $[T, 2T), [2T, 3T), \ldots$, thereby establishing that, at least for $t$ large enough, if $F(x^*, Y(t))$ is small, then $F(x^*, Y(t + h))$ will remains small $\forall h > 0$. As it turns out, this means that after long enough time, if $x_t$ ever visits $\tilde{B}(x^*, \varepsilon)$, it will (almost surely) be forever trapped inside the neighborhood twice that size (i.e. $\tilde{B}(x^*, 2\varepsilon)$). Since Theorem 3.2 ensures that $x_t$ visits $\tilde{B}(x^*, \varepsilon)$ infinitively often (almost surely), the hypothesis is guaranteed to be true. Consequently, this leads to the following claim: $\forall \varepsilon > 0, \exists \tau_0$ (a positive integer), such that:

$$F(x^*, Y(\tau_0 + h)) < \varepsilon, \forall h \in [0, \infty), \quad \text{a.s.}. \tag{4.4}$$

To conclude, Equation (4.4) implies that $F(x^*, Y_t) \to 0$, a.s. as $t \to \infty$, where the SMD iterates $Y_t$ are values at integer time points of the affine trajectory $Y(\tau)$. Per Statement 1 in Lemma 3.1, this leads to that $\|C(Y_t) - x^*\| \to 0$, a.s. as $t \to \infty$, thereby establishing that $X_t = C(Y_t) \to x^*$, a.s..

*Proof.* We prove in turn each of the 6 claims laid out in Remark 4.1.

1. Since $h(\cdot)$ is $K$ strongly convex, by a standard result in convex analysis [5], $C(\cdot)$ is $\frac{1}{K}$-Lipschitz continuous. Since $v(x)$ is locally lipschitz continuous by assumption and bounded since $\mathcal{X}$ is compact, $v(Q(\cdot))$ is locally Lipschitz continuous and bounded. Consequently, standard results in differential equations ([2]) imply that a unique solution exists for the ODE.

2. Connect linearly the SMD iterates $Y_1, Y_2, \ldots, Y_k, \ldots$ at times $0, \alpha_1, \alpha_1 + \alpha_2, \ldots, \sum_{i=0}^{k-1} \alpha_i, \ldots$ yields the following affine curve $Y(t)$:

$$Y(t) = Y_k + (t - \sum_{i=0}^{k-1} \alpha_i) \frac{Y_{k+1} - Y_k}{\alpha_k} \quad \text{for all } t \in [\sum_{i=0}^{k-1} \alpha_i, \sum_{i=0}^{k} \alpha_i), k = 1, 2, \ldots, \tag{4.5}$$

where we adopt the usual convention that $\sum_{i=0}^{-1} \alpha^i = 0$. [1] gives sufficient conditions that ensure a random affine trajectory to be an asymptotic pseudotrajectory of a semiflow almost surely. We shall state one set of sufficient conditions directly in the current context as follows.

If for some $q \geq 2$, the following list of conditions are satisfied:

(a) $\sup_t \mathbf{E}[\|\tilde{\xi}^{t+1}\|_*^q] < \infty$.
(b) $\sum_{n=0}^{\infty} (\alpha_t)^{1+\frac{q}{2}} < \infty$.
(c) $\sup_t \|x_t\| < \infty$.

Then the affinely interpolated process $Y(t)$ is an asymptotic pseudotrajectory of the semi-flow $\Phi$ induced by the ODE almost surely:

$$\forall T > 0, \lim_{t\to\infty} \sup_{0 \le h \le T} \|Y(t+h), \Phi_h(Y(t))\|_* = 0, \text{a.s.}.$$

Choose $q = 2$, the above conditions can be easily verified: (a) was already verified in the proof to Theorem 3.2; (b) holds since $\alpha_t$ is square summable; (c) holds since the decision space $\mathcal{X}$ is compact. Therefore the claim follows.

3. By a well-known result in variational analysis ([6]), each $h(\cdot)$ is differentiable and

$$\frac{dh^*(y)}{dy} = C(y). \tag{4.6}$$

Note further that since $\Phi_t(y)$ is the solution to the ODE (under the initial condition $y$), we have $\frac{d\Phi_t(y)}{dt} = v(x(t))$. We can thus compute the derivative of Fenchel coupling as follows:

$$\frac{dF(x^*, \Phi_t(y))}{dt} = \frac{d}{dt}\{h(x^*) - \langle \Phi_t(y), x^* \rangle + h^*(\Phi_t(y))\} \tag{4.7}$$

$$= \langle -\frac{d\Phi_t(y)}{dt}, x^* \rangle + \langle C(\Phi_t(y)), \frac{d\Phi_t(y)}{dt} \rangle \tag{4.8}$$

$$= \langle -v(x(t)), x^* \rangle + \langle v(x(t)), C(\Phi_t(y)) \rangle \tag{4.9}$$

$$= \langle -v(x(t)), x^* \rangle + \langle v(x(t)), x(t) \rangle \tag{4.10}$$

$$= \langle v(x(t)), x(t) - x^* \rangle \le 0, \tag{4.11}$$

where the second equality follows from Equation (4.6), and the last inequality follows from $x^*$ is variationally coherent. The monotonicity property therefore follows.

4. For any given $\varepsilon > 0$, pick an $\hat{\varepsilon} > 0$ such that $B(x^*, \hat{\varepsilon}) \subset \tilde{B}(x^*, \frac{\varepsilon}{2})$ (per 3.28). By Equation (4.11), we have

$$\frac{dF(x^*, \Phi_t(y))}{dt} = \langle v(x(t)), x(t) - x^* \rangle < 0, \forall x(t) \ne x^*.$$

Since $\mathcal{X} - B(x^*, \hat{\varepsilon})$ is a compact set and $v(\cdot)$ is a continuous function, we have

$$\langle v(x(t)), x(t) - x^* \rangle \le -a_{\hat{\varepsilon}}, \forall x(t) \in \mathcal{X} - B(x^*, \hat{\varepsilon}), \tag{4.12}$$

for some positive constant $a_{\hat{\varepsilon}}$.

Starting at $y$, by time $s$, there are two possibilities. The first possibility is that $x(s) \in \tilde{B}(x^*, \frac{\varepsilon}{2})$. In this case, by definition,

$$F(x^*, \Phi_s(y)) < \frac{\varepsilon}{2}. \tag{4.13}$$

The second possibility is that $x(s) \notin \tilde{B}(x^*, \frac{\varepsilon}{2})$. This implies that $x(t) \notin B(x^*, \hat{\varepsilon}), \forall t \in [0, s]$, because otherwise, since $B(x^*, \hat{\varepsilon}) \subset \tilde{B}(x^*, \frac{\varepsilon}{2})$, it must be that $x(s_0) \in \tilde{B}(x^*, \frac{\varepsilon}{2})$ for some $s_0 \in [0, s]$. This then implies that, by the monotonicity property established in Claim 3, $F(x^*, \Phi_s(y)) \le F(x^*, \Phi_{s_0}(y))$, thereby leading to $x(s) \in \tilde{B}(x^*, \frac{\varepsilon}{2})$, a contradiction.

Since $x(t) \notin \tilde{B}(x^*, \frac{\varepsilon}{2}), \forall t \in [0, s]$, we have $x(t) \notin B(x^*, \hat{\varepsilon}), \forall t \in [0, s]$, leading to that Equation 4.12 holds for $t \in [0, s]$. Therefore, taking $s = \frac{\varepsilon}{2a_{\hat{\varepsilon}}}$, we obtain:

$$F(x^*, \Phi_s(y)) \le F(x^*, y) - a_{\hat{\varepsilon}}s = F(x^*, y) - \frac{\varepsilon}{2}. \tag{4.14}$$

Equation (4.13) and Equation (4.14) together establish that:

$$F(x^*, \Phi_s(y)) \le \max\{\frac{\varepsilon}{2}, F(x^*, y) - \frac{\varepsilon}{2}\}.$$

5. Let $R = \sup_{x \in \mathcal{X}} \|x\|$, which is finite since $\mathcal{X}$ is compact. By the definition of dual norm:

$$\langle Y(t+h) - \Phi_h(Y(t+h)), C(\Phi_h(Y(t+h))) - x^* \rangle \leq \qquad (4.15)$$

$$\|(Y(t+h) - \Phi_h(Y(t+h))\|_* \|C(\Phi_h(Y(t+h))) - x^*)\| \leq R\|Y(t+h) - \Phi_h(t+h)\|^*. \qquad (4.16)$$

Fix some $T > 0$ and define $K_\lambda = \max_i \frac{\lambda_i}{K_i}$ and $\delta = K(\sqrt{R^2 + \frac{\varepsilon}{K}} - R)$. Per Claim 3, we have

$$\forall T > 0, \lim_{t \to \infty} \sup_{0 \leq h \leq T} \|Y(t+h), \Phi_h(Y(t))\|_* = 0, \text{a.s.}.$$

Consequently, choose $\tau(\delta, T)$ such that $\|Y(t+h) - \Phi_h(Y(t))\|_* < \delta, \forall t \geq \tau$. Expanding Fenchel coupling, we obtain that $\forall t \geq \tau$ and $\forall h \in [0, T]$:

$$F(x^*, Y(t+h)) = F(x^*, \Phi_h(Y(t)) + Y(t+h) - \Phi_h(Y(t))) \leq F(x^*, \Phi_h(Y(t)))$$

$$(4.17)$$

$$+ \langle Y(t+h) - \Phi_h(Y(t)), C(\Phi_h(Y(t))) - x^* \rangle + \frac{1}{2K}\|Y(t+h) - \Phi_h(Y(t))\|_*^2 \quad (4.18)$$

$$\leq F(x^*, \Phi_h(Y(t))) + R\|Y(t+h) - \Phi_h(t)\|_* + \frac{1}{2K}\|Y(t+h) - \Phi_h(Y(t))\|_*^2 \quad (4.19)$$

$$\leq F(x^*, \Phi_h(Y(t))) + R\delta + \frac{1}{2K}\delta^2 \qquad (4.20)$$

$$\leq F(x^*, \Phi_h(Y(t))) + RK(\sqrt{R^2 + \frac{\varepsilon}{K}} - R) + \frac{1}{2K}(K(\sqrt{R^2 + \frac{\varepsilon}{K}} - R))^2 \qquad (4.21)$$

$$= F(x^*, \Phi_h(Y(t))) + \frac{\varepsilon}{2}, \qquad (4.22)$$

where the first inequality follows from Equation (4.16) and the last equality follows from straightforward algebraic verification. The claim is therefore established.

6. We start by fixing an arbitrary $\varepsilon > 0$. Per Claim 4, there exists an $s > 0$ (depending on $\varepsilon$) such that Equation (4.2) holds. Set the horizon $T = s$. Per Claim 5, there exists a $\tau$ (depending on both $\varepsilon$ and $T$) such that Equation (4.3) holds $\forall t \geq \tau$. Now, per Theorem 3.2, $x_t$ visits $\tilde{B}(x^*, \delta)$ infinitely often[3]. Therefore, pick an integer $\tau_0 \geq \tau$ such that $x^{\tau_0} \in \tilde{B}(x^*, \frac{\varepsilon}{2})$. With this choice of $\tau_0$, we know that by definition of $\tilde{B}$,

$$F(x^*, Y(\tau_0)) < \frac{\varepsilon}{2}. \qquad (4.23)$$

Our goal is to establish that $F(x^*, Y(\tau_0 + h)) < \varepsilon, \forall h \in [0, \infty)$. To that end, partition the time $[0, \infty)$ into disjoint time intervals $[0, T), [T, 2T), \ldots, [nT, (n+1)T), \ldots$.

Per Claim 3, the monotonicity property given in Equation (4.1) implies that:

$$F(x^*, \Phi_h(Y(\tau_0))) \leq F(x^*, \Phi_0(Y(\tau_0))) = F(x^*, Y(\tau_0)) < \frac{\varepsilon}{2}, \forall h \geq 0, \qquad (4.24)$$

where the equality follows from the semi-group property of a semiflow.

Per Equation (4.3), for $h \in [0, T)$, we then have:

$$F(x^*, Y(\tau_0 + h)) < F(x^*, \Phi_h(Y(\tau_0))) + \frac{\varepsilon}{2} < \frac{\varepsilon}{2} + \frac{\varepsilon}{2} = \varepsilon, \qquad (4.25)$$

where the last inequality follows from Equation (4.24).

Now assume inductively that Equation (4.25) holds for every $h \in [nT, (n+1)T)$, where $n$ is a non-negative integer. We then have $\forall h \in [nT, (n+1)T)$:

$$F(x^*, Y(\tau_0 + T + h)) < F(x^*, \Phi_T(Y(\tau_0 + h))) + \frac{\varepsilon}{2} \qquad (4.26)$$

$$\leq \max\{\frac{\varepsilon}{2}, F(x^*, Y(\tau_0 + h)) - \frac{\varepsilon}{2}\} + \frac{\varepsilon}{2} \leq \frac{\varepsilon}{2} + \frac{\varepsilon}{2} = \varepsilon, \qquad (4.27)$$

where the first inequality follows from Equation (4.3), the second inequality follows from Equation (4.2), and the third inequality follows from the induction hypothesis $F(x^*, Y(\tau_0 + h)) < \varepsilon$. Consequently, Equation (4.25) holds for every $h \in [(n+1)T, (n+2)T)$, thereby completing the induction and establishing that:

$$F(x^*, Y(\tau_0 + h)) < \varepsilon, \forall h \in [0, \infty).$$

$\square$

**Theorem 4.2.** *Let Opt be a $c$-strongly variationally coherent optimization problem. Then* $\|\bar{x}_T - x^*\|^2 \leq \frac{2}{c} \frac{F(x^*, y_0) + \frac{B}{2K} \sum_{n=0}^{T} \gamma_n^2}{\sum_{n=0}^{T} \gamma_n}$, *where* $\bar{x}_T = \frac{\sum_{n=0}^{T} \gamma_n x_n}{\sum_{n=0}^{T} \gamma_n}$, $K$ *is the strong convexity coefficient of $h$ and $B = \max_{x \in \mathcal{X}} \|\nabla g(x)\|_*^2$.*

*Proof.* First, with some algebra, one can show the following descent inequality of Fenchel coupling: let $y^+ = y + \gamma v$ and $x = Q(y), x^+ = Q(y^+)$, then $\forall p \in \mathcal{X}, \langle v, x - p \rangle \leq \frac{F(p,y) - F(p,y^+)}{\gamma} + \frac{\gamma \|v\|_*^2}{2K}$, where $\| \cdot \|_*$ is the dual norm of $\| \cdot \|$ and $K$ is the strong convexity coefficient of the regularizer $h$. Next, applying the descent inequality to mirror descent, we obtain $\gamma_n \langle \nabla g(x_n), x_n - x^* \rangle \leq F(x^*, y_n) - F(x^*, y_{n+1}) + \frac{\gamma_n^2 \|\nabla g(x_n)\|_*^2}{2K}$. Telescoping then yields $\sum_{n=0}^{T} \gamma_n \langle \nabla g(x_n), x_n - x^* \rangle \leq F(x^*, y_0) - F(x^*, y_{T+1}) + \sum_{n=0}^{T} \frac{\gamma_n^2 \|\nabla g(x_n)\|_*^2}{2K} \leq F(x^*, y_0) + \sum_{n=0}^{T} \frac{\gamma_n^2 B}{2K}$. Finally, $\|\bar{x}_T - x^*\|^2 = \|\sum_{n=0}^{T} \frac{\gamma_n}{\sum_{n=0}^{T} \gamma_n} x_n - x^*\|^2 \leq \frac{\sum_{n=0}^{T} \gamma_n \|x_n - x^*\|^2}{\sum_{n=0}^{T} \gamma_n} \leq \frac{2}{c} \frac{\sum_{n=0}^{T} \gamma_n \langle \nabla g(x_n), x_n - x^* \rangle}{\sum_{n=0}^{T} \gamma_n} \leq \frac{2}{c} \frac{F(x^*, y_0) + \frac{B}{2K} \sum_{n=0}^{T} \gamma_n^2}{\sum_{n=0}^{T} \gamma_n}$. $\square$

## Footnotes

[1]In the above, gradients are treated as elements of the dual space $\mathcal{V}^*$ of $\mathcal{V}$ and $\|v\|_* = \sup\{\langle v, x \rangle : \|x\| \le 1\}$ denotes the dual norm of $v \in \mathcal{V}^*$. We also note that $\nabla G(x; \xi)$ refers to the gradient of $G(x; \xi)$ with respect to $x$; since $\Omega$ need not have a differential structure, there is no danger of confusion.

[2]A crucial point to note is that since $C$ may not be invertible, there may not exist a unique solution for $x(t)$.

[3]All the statements made here are true almost surely. We will omit repeatedly saying that for convenience. Alternatively, one can think of this as a path-by-path argument and each ensuing statement is made for a particular sample path that comes from a probability 1 space.