[Reviews · NeurIPS 2017]

Reviewer 1



This paper analyzes the stochastic mirror descent (SMD) algorithm for a specific class of non-convex functions the variational coherence (VC) assumption. The authors study mirror descent by establishing a correspondence between the SMD iterates to an ordinary differential equation. They introduce a new divergence measure named Fenchel coupling and exploit its monotonicity property (due to the VC assumption) to demonstrate the SMD algorithm decreases its objective value until it gets close to the optimum. The paper is clearly written and - omitting certain details -the proof is relatively easy to follow. The paper however has some significant problems in its presentation: 1) Interest for machine learning community: The main contribution of this paper is a proof of asymptotic convergence of SMD for certain non-convex functions. Although this is an interesting result for the optimization community, I’m however not sure this will be of a broad interest for the ML community which is typically more interested in deriving convergence rates. The authors do not give examples of ML problems where this new analysis could be relevant. There is also no experiment results. Can you please try to address this issue in the rebuttal? 2) Related work: The paper does not cover existing work in optimization for non-convex functions, not does it cover prior work relating ODEs to numerical optimization. I think existing proofs of convergence for stochastic gradient descent and proximal methods should be mentioned, e.g. Ghadimi, Saeed, and Guanghui Lan. "Stochastic first-and zeroth-order methods for nonconvex stochastic programming." SIAM Journal on Optimization 23.4 (2013): 2341-2368. For ODEs, see references in [18] as well as Stochastic Approximation and Recursive Algorithms and Applications Kushner, Harold, Yin, George I’m sure the authors are familiar with this line of work so please cite it appropriately. Fenchel coupling The paper uses the Fenchel coupling to quantify the distance between primal and dual variables thereby serving as an energy function to prove convergence. Can the authors comment on the difficulty of directly using the discrete primal variable to prove convergence? e.g. using a proof similar to SGD for non-convex functions (Stochastic First- and Zeroth-order Methods for Nonconvex Stochastic Programming, Saeed Ghadimi, Guanghui Lan) Can the authors comment on whether other types of divergences would be appropriate for the analysis? It seems to me that the monotonicity property is due to the variational coherence assumption (as can be seen in E 3.39 in the appendix), can you confirm this? If so, then I wonder if the role of the Fenchel coupling is really that important for the analysis to hold? Convergence rate: As mentioned earlier, the ML community would probably be more interested in a convergence rate. Can the authors elaborate on the difficulty of establishing a rate using their ODE technique? Theorem 3.4: This result was at first puzzling to me. It says that the SMD algorithm oscillates in a region near the optimum. This is known to happen for SGD with constant step-size while using a decreasing step size (+ assumptions mentioned in the paper) will then guarantee convergence to an optimum. Can the authors clarify this? Analysis: Page 5 appendix above line 92, should be sum_k \alpha_k = + \infty Line 174: you are citing a book, could you give a more precise reference? -------------- Post-rebuttal: I find the answers in the rebuttal convincing and updated my recommendation score accordingly. Overall, I think the paper would benefit from some clarifications which I think is easily do-able between now and the camera ready version. This includes: 1) Importance of the Fenchel coupling. The use of the Fenchel coupling (instead of the commonly used Bregman divergence) is a nice trick to get convergence for the type of non-convex functions considered in the paper. Also the ODE technique used in this paper could potentially push the development of similar techniques for different optimization techniques so this could really be valuable for the ML community. The submitted version of the paper did not explain this well but the answer provided in the rebuttal is convincing. I would recommend developing this aspect even more in the revised version. 2) Interest regarding the ML community, they should develop this more in the introduction. Adding experimental results as they promised to R3 would be valuable as well 3) As R2 pointed out, the intuition behind the analysis is not always clear. Given the rather convincing answers in the rebuttal, I think the authors can easily improve this aspect in the revised version. 4) Minor modifications: Fix the numbering and clarify the use of certain terms whose meaning might differ for different communities, e.g. linear decrease, last iterate.

Reviewer 2



Review: Stochastic Mirror Descent for Non-Convex Optimization This paper discusses applying stochastic mirror descent (SMD) method on a class of problems beyond traditional convex ones. The authors show that global (non-ergodic) convergence of the SMD is possible for a class of problems called Variationally Coherent problems. The authors have also derived convergence rates, as well as analyzed the local behavior of the SMD for this problems. The paper contains lots of derivations. My general impression is that too many results are packed in the paper, and many of them are stated in a rather vague manner so the main points/contributions are somewhat made obscure. I have a few general concerns about the work. 1) The title of the paper is misleading. The authors should be very specific and say “Stochastic Mirror Descent for Variationally Coherent Optimization Problems”. Claiming what the paper is doing is to perform SMD on general non-convex problems is a stretch. 2) In the introduction and perhaps in the abstract, the authors need to be more precise on what they meant by the “last-iterate”. This could be very misleading since the authors are not actually proving that the “last-iterate” of the algorithm to the global/local mean. 3) Before Eq. (3.3), the authors mentioned that F(x^*,\Phi_t) will decrease no slower than linearly, however I don’t see where such a “linear” decrease is in Eq. (3.3). 4) It is not clear from the main part of the paper what is the intuition behind Step 2 in page 6, which the reviewer believes is one of the key step of the paper. 5) Please give concrete examples for the locally variationally coherenet (LVC) problem. Also, from the definition of the LVC problem, the (LVC) property is only required to hold in one region. However, it is not clear why the SMD algorithm will be able to reach such a region. Is it possible that SMD will be stuck at some saddle points that are not LVC? From the proof I can say that this is a local result meaning the Algorithm 1 must start inside LVC region, but from the statement of the Theorem it is not clear (the authors simply say that “X_t be the iterate generated by Algorithm 1”). Please clarify. 6) I am not convinced by the result sin Section 5. I think the author should remove it, and use the space to give better results, and possibly some more meaningful derivation, for the other parts of the proof. The authors need to make it more clear about the relationship between problems with sharp minimum, and “generic” linear program. What are other problems that contained in this class? And why the solution of such generic linear program is unique (Corollary 5.3)? Also from the proof of Theorem 5.4, it appears that the authors call Theorem 3.3 (or Theorem 3.4 in the main paper) to argue for X_t converges to x^* also. However, Theorem 3.3 is based on the fact that the problem is globally (not locally) variationally coherent. But it appears, as the author stated, that the problem considered in Section 5 is locally VC (line 284: … that a sharp minimum is locally VC). 7) There is a mismatch between the theorem number in the supplementary material and that in the main paper. Therefore in the proof when the authors refers to the theorem number sometimes it is confusing.

Reviewer 3



This paper proved that stochastic mirror descent is non-ergodically but asymptotically convergent on a class of non-convex problems named variationally coherent problems. It also showed that stochastic mirror descent is still convergent with high probability if the problem is only locally variationally coherent. I have the following concerns about this paper. First, all the convergence results in this paper are asymptotic. It would be much better if non-asymptotic convergence result with specific convergence rate can be given. Second, in convex optimization, the most common reason to use mirror descent in that it can lead to smaller constant factors in convergence rate. However, this paper failed to justify why mirror descent is still preferred over other common methods like gradient descent in non-convex settings. There is no such theory in this paper due to the lack of convergence rate. Third, no empirical experiments were conducted to show the superiority of this method. So it is not clear whether the proposed method is of practical interests. (Updated after Rebuttal) I read the authors' rebuttal and decided to increase my evaluation.